# Annotation-Efficient Strategy for Segmentation of 3D Body Composition

**Lena Philipp**[1]          Lena.philipp@radboudumc.nl

**Maarten de Rooij**[1]          Maarten.deRooij@radboudumc.nl

**John Hermans**[1]          John.Hermans@radboudumc.nl

**Matthieu Rutten**[1]          Matthieu.Rutten@radboudumc.nl

**Horst Hahn**[2]          horst.hahn@mevis.fraunhofer.de

**Bram van Ginneken**[1]          Bram.vanGinneken@radboudumc.nl

**Alessa Hering**[1,2]          Alessa.Hering@radboudumc.nl

[1] *Department of Medical Imaging, Radboudumc, Nijmegen, The Netherlands*

[2] *Fraunhofer MEVIS, Bremen, Germany*

**Editors:** Accepted for publication at MIDL 2024

## Abstract

Body composition as a diagnostic and prognostic biomarker is gaining importance in various medical fields such as oncology. Therefore, accurate quantification methods are necessary, like analyzing CT images. While several studies introduced deep learning approaches to automatically segment a single slice, quantifying body composition in 3D remains understudied due to the high required annotation effort. This study proposes an annotation-efficient strategy using an iterative self-learning approach with sparse annotations to develop a segmentation model for the abdomen and pelvis, significantly reducing manual annotation needs. The developed model demonstrates outstanding performance with Dice scores for skeletal muscle (SM): 0.97+/-0.01, inter-/intra-muscular adipose tissue (IMAT): 0.83 +/- 0.07, visceral adipose tissue (VAT): 0.94 +/-0.04, and subcutaneous adipose tissue (SAT): 0.98 +/-0.02. A reader study supported these findings, indicating that most cases required negligible to no correction for accurate segmentation for SM, VAT and SAT. The variability in reader evaluations for IMAT underscores the challenge of achieving consensus on its quantification and signals a gap in our understanding of the precision required for accurately assessing this tissue through CT imaging. Moreover, the findings from this study offer advancements in annotation efficiency and present a robust tool for body composition analysis, with potential applications in enhancing diagnostic and prognostic assessments in clinical settings.

**Keywords:** Body composition, 3D, CT, Noisy Annotations, Medical Image Segmentation.

## 1. Introduction

A body mass index (BMI) greater than 30 kg/m2 is commonly seen as a health risk factor for various cardiovascular diseases (Powell-Wiley et al., 2021) and cancer types (Lauby-Secretan et al., 2016). However, the BMI is limited in its inability to differentiate between adipose tissue/muscles and to account for the heterogeneity in fat distribution, which can lead to imprecise or misleading results (Piché et al., 2018). The importance of body composition is increasingly recognized in studying survival in cancer patients (Cespedes Feliciano et al., 2018), (Shachar et al., 2016), implications for care (Prado et al., 2018) and metabolic

diseases (Pi-Sunyer, 2019) and highlights the need for accurate methods of measurement. The widespread use of CT imaging in clinical settings, coupled with its ability to distinguish between adipose tissue and muscle and provide detailed insights into fat distribution, positions it as a superior method for improving body composition analysis accuracy.

In routine clinical practice, measuring body composition remains difficult due to the expert knowledge required for the time-consuming annotation process. This is why only an axial slice at the level of the lumbar spine level 3 (L3) is often used due to the high correlation to the entire volume (Mourtzakis et al., 2008). Convolutional neural networks have enabled the automatic segmentation of skeletal muscle (SM), inter-/intra-muscular adipose tissue (IMAT), visceral adipose tissue (VAT), and subcutaneous adipose tissue (SAT) using a slice at the height of L3 ((Weston et al., 2019), (Paris et al., 2020), (Shen et al., 2023)), L3/L4 ((Nowak et al., 2020), (Park et al., 2020)), pelvis (Hemke et al., 2020), and multiple heights (Ahmad et al., 2023). Additionally, end-to-end solutions have been developed to automate slice selection as well ((Nowak et al., 2021), (Dabiri et al., 2020), (Zhang et al., 2022a), (Bridge et al., 2022), (Xu et al., 2022)). The trend has recently evolved towards volumetric body composition analysis for more comprehensive measurements across larger body regions using multi-atlas segmentation ((Hu et al., 2018), (Decazes et al., 2019)) or deep learning (DL) approaches ((Koitka et al., 2021), (Pu et al., 2021), (Liu et al., 2020), (Dai et al., 2024), (Fu et al., 2020), (Borrelli et al., 2021),(Lee et al., 2021)). Despite the advancements, the annotation of data required for training these 3D DL models presents significant challenges, attributed to the labor-intensive and meticulous nature of the task. This often results in training and test sets that are either small in size or only partially annotated. (Lee et al., 2021) and (Pu et al., 2021) utilized larger training datasets to enhance their models. (Lee et al., 2021) employed manual annotations for their training and testing sets, leading to a model that achieved high Dice scores. However, it did not distinguish IMAT with a separate segmentation mask. Addressing the complexity of the volumetric annotation of body composition, (Pu et al., 2021) tackled this challenge by employing semi-supervised self-training in the training process, albeit with suboptimal performance outcomes with Dice scores of 0.82 for VAT and 0.59 for IMAT.

Semi-supervised learning is particularly relevant in contexts where extensive data exists, but only a fraction is labeled. It incorporates not only fully annotated data but also utilizes noisy, weak annotations or pseudo labels to enrich the training process. Pseudo labels are annotations created by applying a model to unlabeled data and using the output as new additional annotations. This self-training step is repeated to extend the training data iteratively or improve the label's quality. Self-training can be seen as a form of weak supervision and builds on the potential to outperform the teacher ((Guan et al., 2018), (Khoreva et al., 2017), (Zhang et al., 2018)). While many 3D semi-supervised segmentation methods require an initial dataset of 3D annotations, utilizing sparse annotations can significantly enhance efficiency by annotating just a few slices within a 3D volume, taking advantage of the strong correlation among slices to preserve precise object boundaries with minimal annotations. Approaches using sparse annotations have been shown to outperform traditional weakly supervised techniques that utilize scribbles ((Lin et al., 2016), (Liu et al., 2022)) and bounding boxes ((Oh et al., 2021), (Dai et al., 2015)) in both efficiency and accuracy. Most of these methods involve the use of registration modules to generate pseudo labels ((Li et al., 2022), (Cai et al., 2023a), (Bitarafan et al., 2020)). (Cai et al., 2023b) propose

a cross-teaching method that enforces consistency between the predictions of 3D and 2D networks, thereby increasing the view difference of networks.

In this work, we aimed to develop a robust 3D segmentation model to quantify SM, IMAT, VAT and SAT. Facing challenges like limited annotated datasets, time-consuming manual labeling, and the need for diverse scans for robustness, we introduce a novel self-training strategy with sparse annotations. This strategy transitions from a 2D to a 3D model, substantially reducing the manual annotation workload by requiring only individual slices to be labeled manually. The goal of this method is to decrease annotation efforts without compromising on segmentation quality. To validate the efficacy of our approach, we conducted an evaluation on an internal test set, focusing on the segmentation performance through the Dice score. Additionally, an expert reader study using a larger external dataset was conducted to evaluate the effort needed to correct generated segmentation masks, offering a detailed assessment of our model's clinical utility and accuracy.

## 2. Materials and Methods

The objective is to create a DL model that accurately measures body composition throughout the abdominal and pelvic regions. This involves training the model to produce segmentation masks for four categories: SM, IMAT, VAT, and SAT.

### 2.1. Data

Our **training dataset** comprises 116 scans, amounting to 38,002 slices, from 100 patients (47% female), gathered between 2008 and 2021. These scans were selected for their comprehensive field of view spanning the entire abdomen and pelvis. A subset of 417 slices (extracted from 24 scans) from this dataset were annotated by a trained researcher and student using an open-source tool 3D Slicer (Slicer, 2020), (Fedorov et al., 2012). For quality control, an experienced radiologist was consulted. The annotation process followed a standardized annotation protocol (Alberta Protocol; (TomoVision, 2021)). The proposed segmentation strategy was applied to the remaining 92 scans and slices. The **internal test set**, aimed at assessing segmentation performance measured as Dice score, includes 20 scans (10 male, 10 female) chosen for their field of view. Given the labor-intensive nature of annotating each of these scans (each with an average of 322 slices) this set was initially automatically annotated using an intermediate baseline 3D U-Net trained on 12 CT scans. These automatically derived segmentation masks were subsequently manually verified and corrected. For **external testset for visual assessment**, we utilized 100 cases from the KiTS21 dataset (Heller et al., 2023), ensuring a balance in BMI categories (normal, overweight, obese) and sex, using the KiC data (IBM Research, 2022). These cases were adjusted to focus on the L1 to L5 region and resampled to a 3mm slice thickness to streamline the review process. Further data details are provided in Appendix A.

### 2.2. Efficient Annotation and Training Strategy

The proposed annotation strategy employs a self-training methodology, utilizing both 2D and 3D neural networks.

**2D training:** The process began with the selection of a subset of scans, totaling 10.

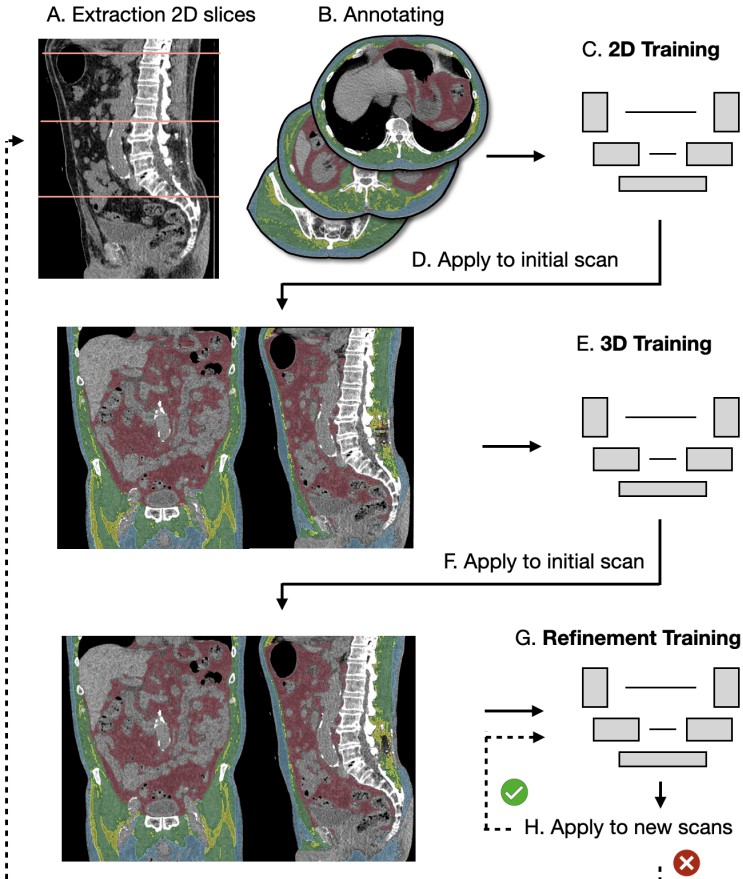

Figure 1: Efficient Annotation and Training Workflow. 2D slices are extracted (A) and annotated as training data (B). A 2D U-Net model is trained on these slices (C) to generate masks for all slices of the scan (D). The 2D model's output is assembled into 3D masks. These masks are used to train a 3D model (E), which generates refined segmentation masks (F). These 3D masks guide the training of a second 3D model for enhanced accuracy (G). Iterative retraining with additional targeted scans refines the model further, masks and scans are either used directly as a training example or they are used to retrain the 2D model (H).

For each scan, 15-20 slices were chosen, spanning from the T9 vertebra to the end of the pelvis, and were semi-automatically annotated using a threshold brush to create training data. A 2D U-Net model was then specifically trained to segment these slices accurately. To keep the model's training focused, we avoided using regularization techniques like data augmentation or dropout. After training, this 2D model was used to produce segmentation masks for all slices in the scans, which were then assembled into complete 3D masks.

**3D training:** The initial, imperfect masks served as the training foundation for a 3D nnU-Net (Isensee et al., 2021). Leveraging the added contextual patterns, this 3D model was adept at generating refined, smoother masks for the scans.

Table 1: Overview of the full model's performance on the internal test set. The rows present the Dice scores per class averaged across all scans in the corresponding sections.

| Section | SM | IMAT | VAT | SAT | Mean |
|---|---|---|---|---|---|
| Top | $0.952 \pm 0.02$ | $0.833 \pm 0.05$ | $0.919 \pm 0.04$ | $0.974 \pm 0.03$ | $0.919 \pm 0.07$ |
| L1 - L5 | $0.969 \pm 0.01$ | $0.834 \pm 0.04$ | $0.948 \pm 0.04$ | $0.987 \pm 0.01$ | $0.935 \pm 0.07$ |
| Pelvis | $0.98 \pm 0.005$ | $0.861 \pm 0.04$ | $0.932 \pm 0.03$ | $0.982 \pm 0.01$ | $0.939 \pm 0.06$ |
| Mean | $0.973 \pm 0.007$ | $0.848 \pm 0.04$ | $0.944 \pm 0.04$ | $0.983 \pm 0.01$ | $0.937 \pm 0.06$ |

**Refinement Process:** After the initial prediction phase, a second nnU-Net was trained using the improved masks from the first 3D U-Net to enhance segmentation accuracy further. This second model is applied to segment new scans. If this model encounters difficulties or inaccurately segments a scan, as determined by human visual assessment, an iterative refinement process is initiated. Starting with step one, slices for the failure case are extracted and used for retraining the 2D U-Net to improve the 3D model's accuracy.

This process was repeated three times. Each time the pool of training slices was extended with slices from a focused scan group (certain kernel types, low dose CT scans, different age groups). With this strategy, expert annotations were only performed for a small subset of 2D masks (417 out of 38002, 1%) instead of correcting 3D masks, which significantly lower the overall annotation workload.

### 2.3. Evaluation

Our evaluation comprises two phases to ensure a thorough assessment of the model:
**Quantitative Assessment:** We utilize a dataset of 20 cases to measure the model's Dice score performance, focusing on the area from the thoracic vertebra T9 to the end of the pelvis which we divided into three specific segments: T9-T12, L1-L5, and S1 to the end of the pelvis. **Visual Assessment:** Acknowledging the limitations presented by the small size of our internal test set, we incorporated an additional visual evaluation step to provide a broader perspective on the model's performance. Three experienced radiologists visually evaluate 100 cases within the L1-L5 region, rating the effort to correct the segmentation masks. With this evaluation, conducted on the grand-challenge platform (grand-challenge.org, 2024), we gain insights into the model's current performance and usability. The rating scale, adapted from (Berta et al., 2021), ranges from 1 (extensive effort) to 5 (no effort). Further details on this scale are available in the Appendix B.

### 3. Results

**Dice Scores per Class and Segment.** The final 3D model was evaluated using the internal test set. To see the performance for different parts of the body, the performance with split into different segments. The model achieved high precision across all sections, reflected in the high Dice scores for SM, VAT, and SAT, with slightly more variability noted in the IMAT segmentations. The results are presented in Table 1. For additional details, refer to the boxplot in Appendix C and the visual overview of a sample case in Appendix D.

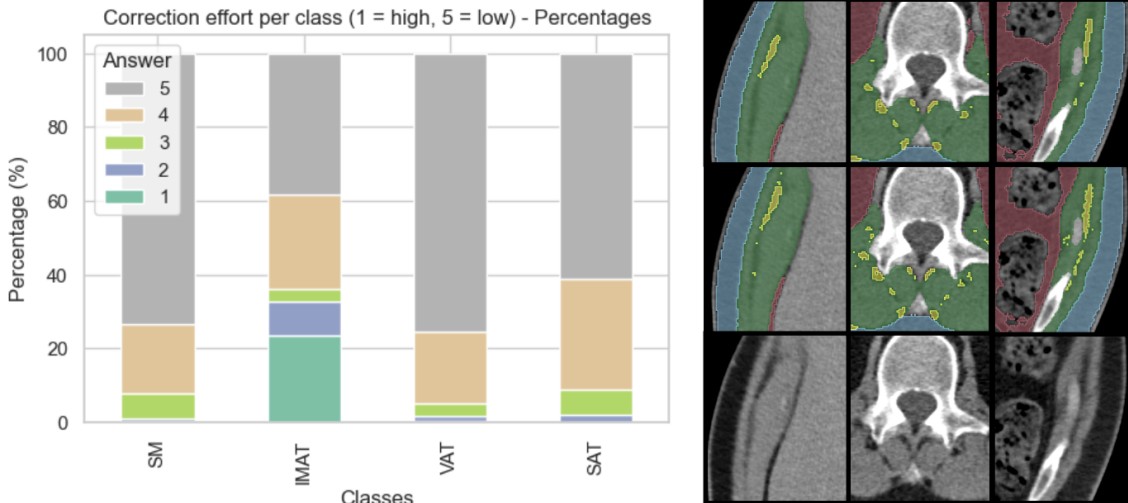

Figure 2: Left: Aggregated answers of all readers for the 100 cases across classes. Right: An illustrative case from the reader study, displaying segmentation of IMAT where two readers scored a 5, and one reader gave a score of 1. Row 1: Automatically generated mask. Row 2: Manually corrected mask. Row 3: Original image.

**Reader Study.** The aggregated scores from the reader study (Figure 2) indicate that most cases required minimal to no corrections for SM, VAT, and SAT, highlighting the model's high accuracy and reliability for these classes. Specifically, for SM, all readers assigned a score of at least 4 in 81 cases, denoting negligible to no correction needed, and in 97 cases, the score was above 3, suggesting limited correction effort. Similar results are obtained for SAT, for which 79 cases received a score of at least 4, while 94 cases scored above 3. VAT segmentation also demonstrated promising results, with 90 cases rated with a score of 4 or 5, and 97 cases scored above 3. However, IMAT proved more challenging. Two readers found the vast majority (90 cases) to require minimal correction, scoring 4.18 and 4.87 on average, while one reader's average score was 1.32, mainly due to unannotated fatty streaks in muscles, suggesting significant correction time. Figure 2 presents an example of such a case. Correcting this case with a threshold brush and separating all voxels with HU values below -29 from the SM mask resulted in only minor changes. This points to an inherent limitation of CT imaging in differentiating small, closely aligned structures as IMAT (partial volume effects). For more challenges identified in the reader study, see Appendix F.

**Annotation Method.** The annotation method's effectiveness is shown through targeted evaluations, using three manually corrected training dataset examples to track progressive improvement in annotation quality via self-learning, measured by the Dice score. Starting with a foundation of 10 training cases, translating to roughly 100 2D slices, the initial 2D model demonstrated strong performance, especially in segmenting SM and SAT. Training the 3D model with initial masks notable improvement, increasing Dice scores for SM (0.978 to 0.99), IMAT (0.874 to 0.95), VAT (0.963 to 0.989), and SAT (0.989 to 0.99). This

progression underscores the benefits of our phased training approach. The 3D context improved IMAT segmentation by refining inconsistencies across slices, leading to smoother masks. Subsequent application of the 3D model for new mask predictions in these cases resulted in a slight increase in the Dice scores (SM: 0.99, IMAT: 0.977, VAT: 0.994, SAT: 0.997). Upon completion of the third and final iteration, which included 24 cases equivalent to 417 2D slices, the 2D model's Dice scores saw an increase, indicating that a greater variety of training examples were instrumental in boosting the model's performance across the three cases, even without adding more slices of these scans (SM: 0.978, IMAT: 0.892, VAT: 0.965, SAT: 0.99). Visually, the model's progress is evident from increasingly accurate segmentation outcomes on slices from different patients, detailed in the Appendix E.

## 4. Discussion and Conclusion

**Annotation-Efficient Segmentation Approach**. This work introduces a novel annotation-efficient training strategy for 3D body composition segmentation in the abdominal and pelvic cavity, which drastically reduces the need for expert annotations. By self-learning and using 2D and 3D models, only approximately 1% of the slices used in training required manual labeling. The evaluation shows that subsequent training steps effectively reduced the label noise in the 2D model's output. Additionally, incorporating a human-in-the-loop approach for expanding the training dataset further enhanced the final model's performance, demonstrating the strategic value of combining automated refinement with expert human oversight in the training process. The outcome is a robust model capable of producing high-quality segmentation masks, as evidenced by the evaluations.

However, this approach comes with its challenges. While self-training is particularly effective for small labeled datasets (Bai et al., 2017), its advantages are limited for larger datasets. Moreover, adding weak labels introduces the risk of overfitting to noise and adopting incorrect patterns. In this approach, the reliability is roughly assessed by human oversight. Inadequate masks lead to selected scans being redirected into manual single slice annotation, incorporating an element of active learning. Future improvements could involve automating this process through techniques like uncertainty estimation to flag poor examples. It is also valuable to investigate strategies to counteract negative impacts on the learning process, such as modifying loss functions ((Ghosh et al., 2017), (Zhang and Sabuncu, 2018)) or employing sample weighting strategies ((Ren et al., 2018), (Mirikharaji et al., 2019)). Regarding the sparse annotation approach, further promising ideas to explore involve 2.5D approaches for multi-view fusion (Zhang et al., 2022b). Leveraging additional views for sparse annotations introduces more variability and has demonstrated encouraging outcomes (Cai et al., 2023b). Future research should compare semi- and weakly supervised methods evaluated on datasets with wide field of views, such as body composition segmentation, since they present unique challenges.

**Robustness**. Our model is trained on a diverse dataset that includes different slice thicknesses, convolution kernel types, low-dose CT scans, and images from multiple manufacturers, enhancing its adaptability to various image qualities and noise levels, particularly in low slice thickness scenarios.

In evaluating the model, we noted its strong performance in segmenting SM, VAT, and SAT from early development stages, even for challenging tasks like distinguishing VAT as

fat within the abdominal cavity from organ-encased and pericardial fat. However, difficulties arose with both small VAT volumes, complicating SM segmentation near organs, and high SAT or VAT distributions, leading to bigger segmentation gaps. This observation prompted a reevaluation of the training samples to underscore the significance of body type as a critical factor in sample selection. One challenge in creating a balanced dataset stems from the inability to determine fat and muscle distribution from standard scan data or BMI alone. To mitigate this, we propose adopting a sampling strategy that considers demographic characteristics, following the insights from (Magudia et al., 2021) highlighting the variance that comes from age, ethnicity, and sex in automated body composition analysis.

**Inclusion of IMAT.** An interesting addition to our study is the inclusion of IMAT as a separate class, a feature often omitted in other studies due to the complexity and time-consuming nature of its annotation. The Dice score indicates a reasonably high level of precision for IMAT segmentation; however, this metric prompts several questions. The absence of exclusively expert-annotated references could be limiting the Dice score's reliability, a factor that is particularly critical for accurately quantifying the relatively small volume of this class. However, even with references like that available, the reader study underscores concerns about whether voxel-based annotations of IMAT alone on CT images are adequate, highlighted by one reader's assessment that most cases required extensive annotation efforts. Given IMAT's small volume, partial volume effects, image noise, and even minimal inaccuracies could have a disproportionate impact on its measurement. A potential approach to these challenges might involve the use of a unified mask for IMAT and SM, coupled with histogram analysis to evaluate the overall attenuation within the muscle. This could provide insights into muscle quality or the patient's overall health status, although the specific application of such an approach remains to be fully explored.

**Limitations.** Additionally, scans with severe beam hardening or strong truncation artifacts were excluded from the study. Future work, inspired by strategies such as those proposed by (Xu et al., 2023) for managing truncation artifacts on individual slices, is essential. An overview of our model's performance in scenarios complicated by artifacts and other challenging variations is detailed in the Appendix F, offering insights into its current robustness and areas for enhancement.

The dimension of our internal test set is acknowledged as limited, underscoring the necessity for broader validation. A reader study incorporating external datasets has begun to affirm our model's efficacy and robustness, yet it is clear that further validation is needed.

**Conclusion.** This work presents an efficient annotation strategy for 3D body composition segmentation that drastically reduces manual annotations while improving model accuracy via iterative improvements. By training with a diverse dataset and specifically identifying IMAT as a distinct class, our model aims to adapt to various imaging conditions and offers in-depth body composition analysis. The divergent assessments of IMAT by readers not only highlight the difficulties in standardizing its measurement but also point to an essential area for further investigation regarding the accuracy needed for CT imaging-based assessments. Overall, the ability to assess body composition in 3D enhances patient care by enabling precise monitoring of changes, providing deeper insights into patients' health status. Assessing these measurements holds potential for refining outcome predictions and diagnostics, especially when integrated with other health parameters.

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

## Appendix A. Data Details

Table 2: Overview of Patient Characteristics - Training Data.

|  | Male (n = 52) | Female (n = 46) | All (n = 100) |
|---|---|---|---|
| Mean age | $59 \pm 15.1$ | $58.7 \pm 17.2$ | $58.87 \pm 16.02$ |
| 20 - 40 | 7 | 7 | 14 |
| 40 - 65 | 26 | 20 | 46 |
| 65 - 80 | 16 | 17 | 33 |
| >80 | 3 | 2 | 5 |
| NA |  |  | 2 |

Table 3: Overview of Patient Characteristics - Internal Test Data.

|  | Male (n = 10) | Female (n = 10) | All (n = 20) |
|---|---|---|---|
| Mean age | $52.7 \pm 18.3$ | $53 \pm 15.6$ | $53.1 \pm 16.5$ |
| Age range | 28 - 84 | 31 - 79 | 28 - 84 |

Table 4: Overview of Acquisition Parameters - Training Data.

| Parameter | Value |
|---|---|
| Slice thickness range | 0.5 - 5.0 |
| <2 | 61 |
| 2 - 3 | 40 |
| >3 | 15 |
| Manufacturer |  |
| Toshiba | 46 |
| Siemens | 44 |
| Philips | 17 |
| GE Medical Systems | 7 |
| Canon Medical Systems | 2 |
| Contrast - enhanced | 76 |
| Non - contrast - enhanced | 40 |

Table 5: Overview of Acquisition Parameters - Internal Test Data.

| Parameter | Value |
|---|---|
| Slice thickness range (mm) | 1.0 - 3.0 |
| <2 | 10 |
| 2 - 3 | 10 |
| Manufacturer | |
| Toshiba | 17 |
| Siemens | 3 |
| Contrast - enhanced | 19 |
| Non - contrast - enhanced | 1 |

Table 6: Overview of Patient Characteristics - External Test Data.

| | Male (n = 47) | Female (n = 53) | All (n = 100) |
|---|---|---|---|
| Mean age | $59 \pm 13$ | $60.6 \pm 14$ | $60.1 \pm 13.5$ |
| 20 - 40 | 2 | 6 | 8 |
| 40 - 65 | 32 | 25 | 57 |
| 65 - 80 | 9 | 19 | 28 |
| >80 | 4 | 3 | 7 |
| BMI | | | |
| Normal (>18.5 - 25) | 14 | 18 | 32 |
| Overweight (25 - 30) | 16 | 16 | 32 |
| Obese (>30) | 17 | 19 | 36 |

## Appendix B.  Scale Reader Study

1. Extensive effort: segmentation with extensive errors requiring the reader excessive effort to correct them

2. Considerable effort: segmentation with errors that require sizeable and/or time-consuming corrections

3. Limited effort: segmentation with inaccuracies that require limited and/or brief correction

4. Insignificant effort: segmentation with small imperfections negligible for the reader

5. No effort: segmentation corresponding to the ideal result for the reader

## Appendix C. Boxplot Internal Testset

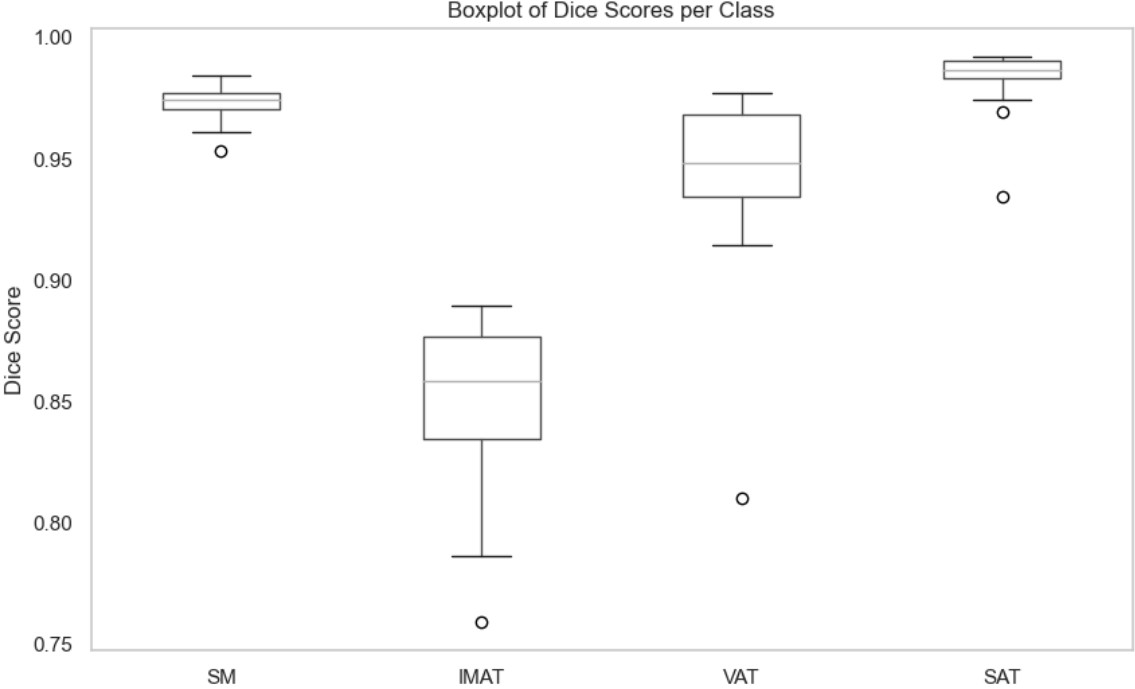

Figure 3: Visualization of the distribution of Dice scores for each class.

Figure 3 illustrates the distribution of Dice scores for each class. Notably, SAT and SM exhibited tight clustering near perfect scores, indicating consistent and accurate segmentation. IMAT and VAT, while also high, showed more variability, suggesting areas where the model may require refinement.

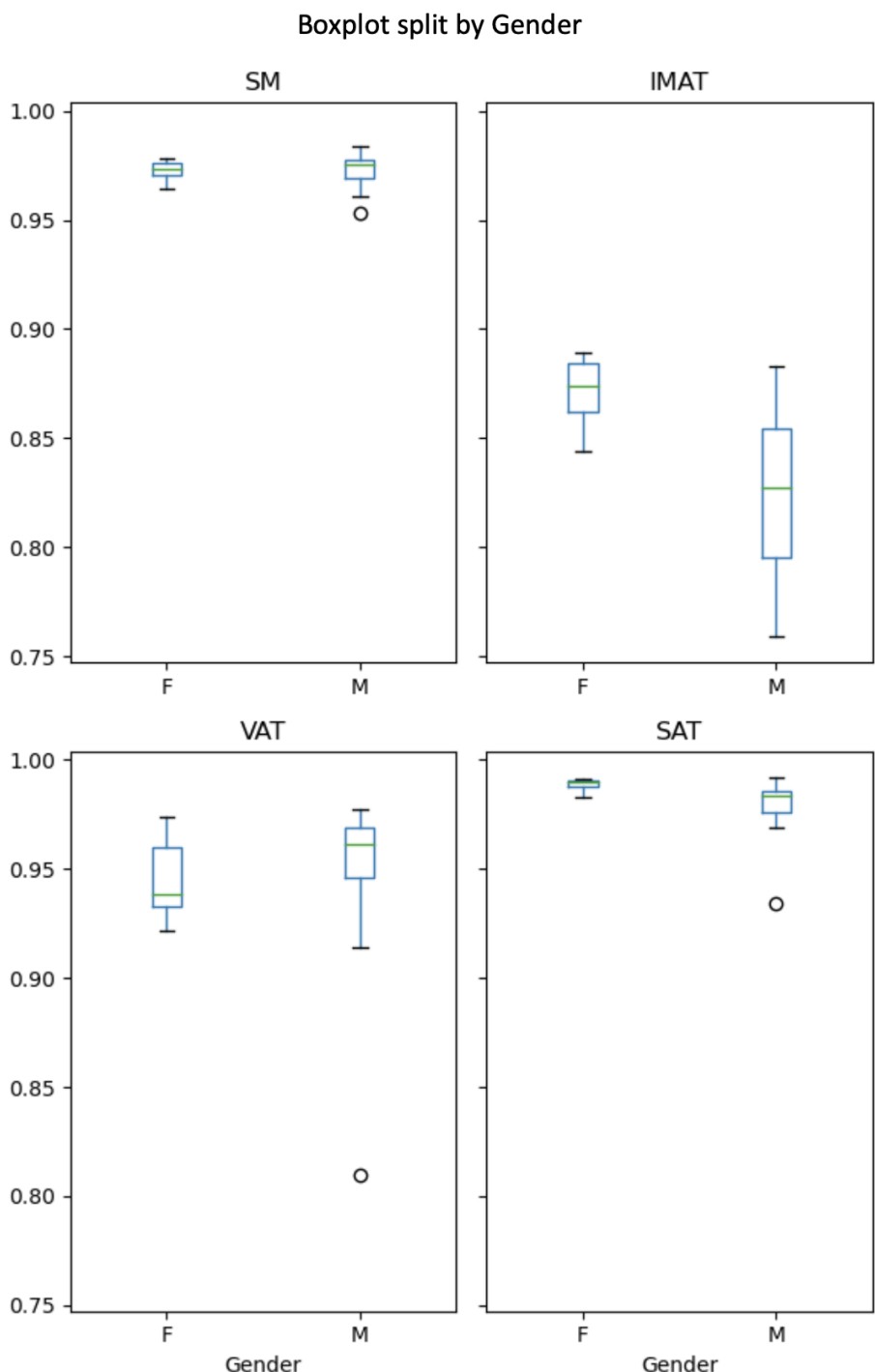

Figure 4: Visualization of the distribution of Dice scores for each class split by gender.

## Appendix D. Visual Overview of Segmentations from Cases of the Internal Testset

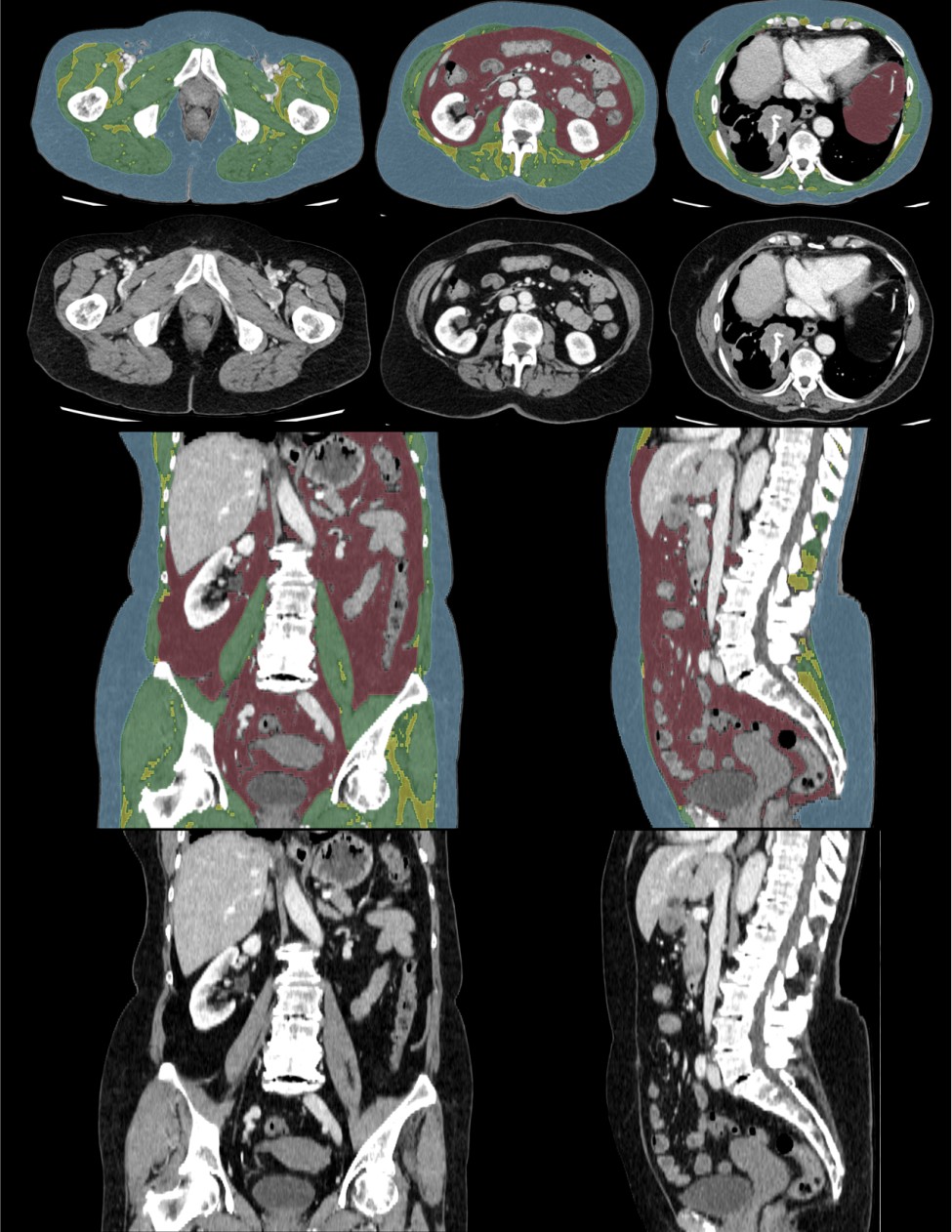

Figure 5: Representative case from the internal test set with segmentation masks generated by the final model. Top row: Segmentation of the first, middle, and last axial slices of the scan. Bottom row: Coronal and sagittal views of the generated masks.

## Appendix E. Visualization Annotation Process

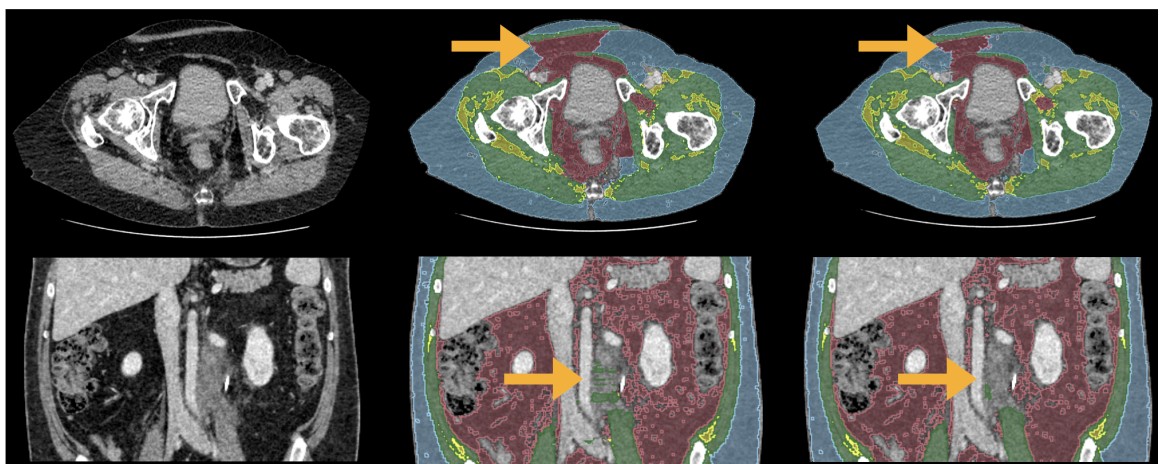

Figure 6: Segmentation output for two training cases from the 2D and 3D model. Column 1: Original image. Column 2: Generated masks by the 2D model. Column 3: Generated masks by the 3D model trained with noisy masks.

## Appendix F.  Failure Cases from Reader Study

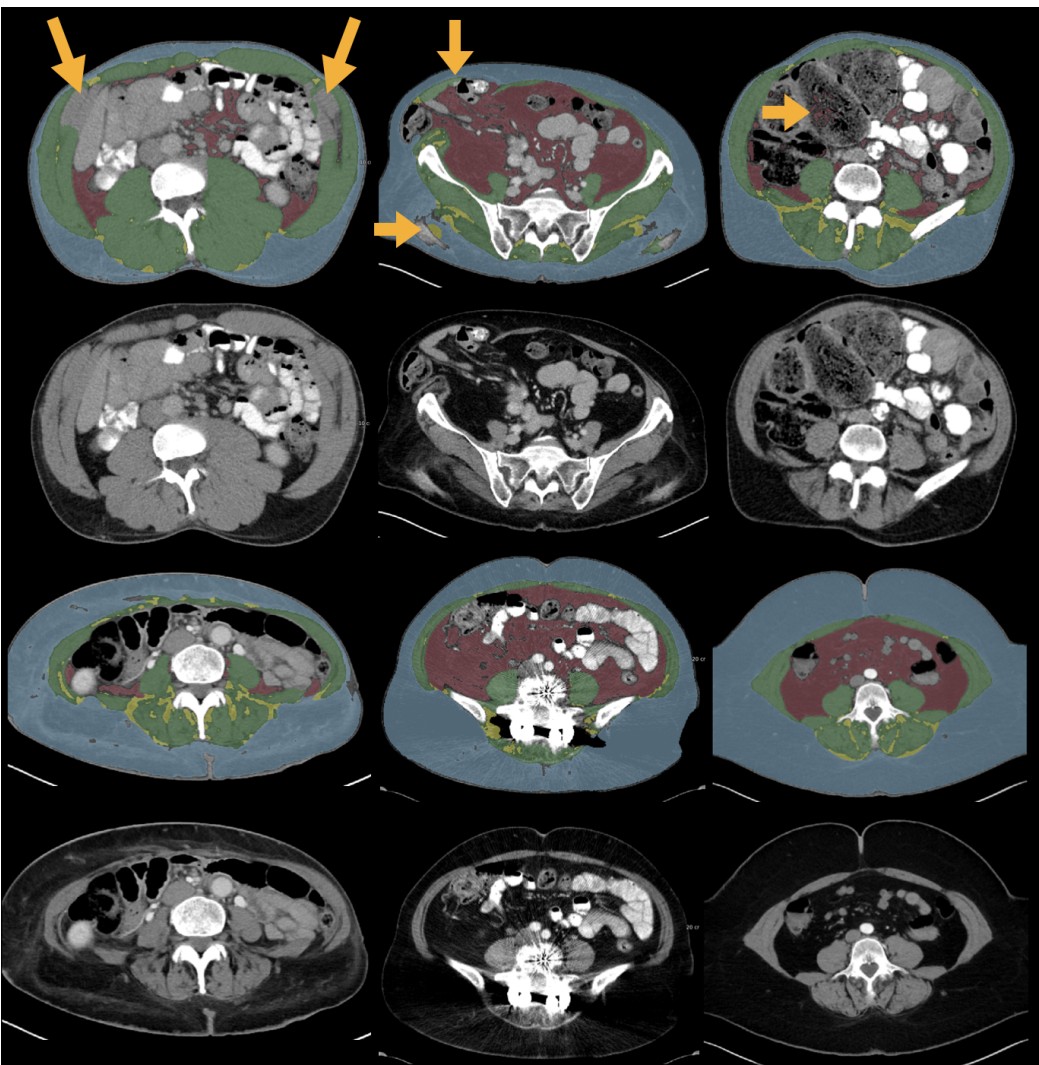

Figure 7: Axial CT slices and corresponding masks illustrating segmentation challenges. These cases received a score of 3 or less from at least one of the readers. Top row, column 1: Undersegmentation of muscle tissue is evident. Top row, column 2: A hernia complicates VAT/SAT differentiation and oversegmentation of IMAT can be seen. Top row, column 3: Misclassification of fat within the bowel as VAT. Bottom row, column 1: Water within the SAT area complicates its quantification. Bottom row, column 2: Scan artifacts lead to distorted predictions. Bottom row, column 3: Truncation of the scan affects SAT estimation accuracy.

