# OpenReview forum: "Annotation-Efficient Strategy for Segmentation of 3D Body Composition"
_MIDL.io/2024/Conference — MIDL 2024 Poster_

### Official Review · Reviewer_KEoR · 2024-02-28

**Confidence:** 5
**Preliminary Rating:** 3
**Final Rating:** 3.5

**Summary:**

In this paper, the authors propose an iterative self-training strategy for annotation-efficient segmentation of body composition for the abdomen and pelvis.  The task focus on 3D medical image segmentation task learning with 2D sparse annotations, including the segmentation of  four categories: SM, IMAT, VAT, and SAT. The proposed method is evaluated on several private and public datasets.

**Strengths:**

The task of annotation-efficient learning for medical image segmentation in novel and interesting. And the paper is generally well-written. The proposed method is evaluated on several private and public datasets.

**Weaknesses:**

The introduction states several annotation-efficient efforts, like learning with noisy, weak annotations or pseudo labels.
Among annotation-efficient approaches, the task in this paper is 3D segmentation with 2D sparse annotations, several related work should be introduced and discussed.

[1] Bitarafan, Adeleh, et al. "3D image segmentation with sparse annotation by self-training and internal registration." IEEE Journal of Biomedical and Health Informatics 25.7 (2020): 2665-2672.
[2] Cai, Heng, et al. "3D Medical Image Segmentation with Sparse Annotation via Cross-Teaching Between 3D and 2D Networks." International Conference on Medical Image Computing and Computer-Assisted Intervention, 2023.
[3] Zhang, Yichi, et al. "Bridging 2D and 3D segmentation networks for computation-efficient volumetric medical image segmentation: An empirical study of 2.5D solutions." Computerized Medical Imaging and Graphics 99 (2022): 102088.

For Kits 21 challenge which focus on kidney tumor segmentation, is there ground truth annotations of targets in this work? Or did the authors annotate corresponding targets. I didn’t see this information in the manuscript.

It would be interesting to compare proposed method with related work stated above, and 2D semi-supervised methods, to further validate the effectiveness of proposed method.

Minor issues: missing reference in p2 paragraph2

**Detailed Comments:**

The introduction states several annotation-efficient efforts, like learning with noisy, weak annotations or pseudo labels.
Among annotation-efficient approaches, the task in this paper is 3D segmentation with 2D sparse annotations, several related work should be introduced and discussed.

[1] Bitarafan, Adeleh, et al. "3D image segmentation with sparse annotation by self-training and internal registration." IEEE Journal of Biomedical and Health Informatics 25.7 (2020): 2665-2672.
[2] Cai, Heng, et al. "3D Medical Image Segmentation with Sparse Annotation via Cross-Teaching Between 3D and 2D Networks." International Conference on Medical Image Computing and Computer-Assisted Intervention, 2023.
[3] Zhang, Yichi, et al. "Bridging 2D and 3D segmentation networks for computation-efficient volumetric medical image segmentation: An empirical study of 2.5D solutions." Computerized Medical Imaging and Graphics 99 (2022): 102088.

For Kits 21 challenge which focus on kidney tumor segmentation, is there ground truth annotations of targets in this work? Or did the authors annotate corresponding targets. I didn’t see this information in the manuscript.

It would be interesting to compare proposed method with related work stated above, and 2D semi-supervised methods, to further validate the effectiveness of proposed method.

Minor issues: missing reference in p2 paragraph2

**Justification Of Final Rating:**

The author's revised version provides a much smoother overview of the overall objectives and related work. The readability of the article has been significantly improved. However, my main concern lies in the fact that this paper does not compare proposed method with state-of-the-art methods. Instead, it only presents a baseline implementation on a new segmentation task. Therefore, I will maintain my initial rating.

**Justification Of The Preliminary Rating:**

The task and idea in novel and interesting. The proposed method is evaluated on several private and public datasets. However, there lacks of discussion and comparison with related work for more comprehensive evaluations.

**Questions To Address In The Rebuttal:**

Please see detailed comments.

---

> ### Author Response · Authors · 2024-03-18
>
> We thank the reviewers for their insightful comments on our paper, which have highlighted its strong aspects and, more crucially, the sections that demand further clarification and improvement which helped us to increase the quality of the manuscript. We address each of their points as follows:
>
> ### *The introduction states several annotation-efficient efforts, like learning with noisy, weak annotations or pseudo labels. Among annotation-efficient approaches, the task in this paper is 3D segmentation with 2D sparse annotations, several related work should be introduced*
>
> We greatly appreciate the feedback regarding the need to introduce and discuss annotation-efficient methods, particularly in the context of 3D segmentation with 2D sparse annotations. This revision not only broadens the scope of our literature review but also strengthens the foundation of our study by showcasing the effectiveness of combining semi- and weakly-supervised segmentation approaches.
> We included a paragraph to the introduction about how sparse annotations can significantly improve efficiency and accuracy by leveraging the correlation among slices, outperforming traditional methods that use scribbles or bounding boxes. It mentions approaches using registration modules to generate pseudo labels ((Li et al., 2022), (Cai et al., 2023a), (Bitarafan et al., 2020)) and introduces a cross-teaching method ((Cai et al., 2023b)) that increases consistency between 3D and 2D network predictions.
>
> ### *… and discussed. It would be interesting to compare proposed method with related work stated above, and 2D semi-supervised methods, to further validate the effectiveness of proposed method.*
>
> The discussion of these approaches raises intriguing possibilities for future research, particularly in the context of segmentation tasks such as body composition for the abdomen and pelvis, which comes with unique challenges. The scans used in this analysis cover an extensive field of view, which encompasses different anatomical structures and therefore high variability. The segmentation problem includes challenging details in classes, like IMAT and VAT. It would be interesting to see how other methods perform confronted with such conditions and what challenges arise.
> Other methods reviewed in the introduction were evaluated on MMWHS (CT, heart substructures) (Cai et al., 2023b), on LiTs (CT, liver tumor), KiTS19 (CT, kidney tumor), LA (MR, left atrium) (Li et al., 2022, Cai et al., 2023a). The closest related validation was performed by (Bitarafan et al., 2020), where they benchmarked their method against other weakly supervised approaches using the Visceral dataset, encompassing various organs and the psoas muscle for segmentation, as well as the CHAOS dataset (CT scans of the kidney, liver, and spleen). Their technique demonstrated significant improvements in segmentation accuracy on sparsely annotated data compared to similar methods. However, the outcomes on the Visceral dataset showed greater variability, potentially attributable to the broader field of view involved in the segmentation tasks.
>
> The goal for our paper was the development of a 3D segmentation model for body composition and we use this method to overcome the challenges that come this it. For further technical validation of our method in comparison to sparse annotation approaches and 2D semi-supervised methods, we would need to annotate more data, (e.g. annotations from different planes as it is required for the method of Cai et al., 2023b). We look forward to further discussions with our peers and potentially collaborating with other researchers to apply these advanced methods to our dataset, despite the current limitations in annotated data.
>
> We added a paragraph to the discussion to highlight future directions:
>
> Further promising idea to explore involves 2.5D approaches for multi-view fusion (Zhang et al., 2022). Leveraging additional views for sparse annotations introduces more variability and has demonstrated encouraging outcomes (Cai et al., 2023b). Future research could focus on comparing semi- and weakly supervised methods on datasets with wide field of views, such as body composition segmentation.
>
> ### *For Kits 21 challenge which focus on kidney tumor segmentation, is there ground truth annotations of targets in this work? Or did the authors annotate corresponding targets. I didn’t see this information in the manuscript.*
>
> We only used the CT scans from the Kits dataset for our reader study, since they are publicly available. We automatically generated masks for these scans using our method and asks experts to assess the quality. No manual labels were created. We decided in favour of this public data set in order to have an additional external data set for evaluation.

---

### Official Review · Reviewer_BTy9 · 2024-02-28

**Confidence:** 4
**Preliminary Rating:** 5

**Summary:**

The authors propose an annotation-efficient strategy for 3D images with a human in the loop. Their strategy requires only a number of slices (2D images) to be segmented initially instead of whole scan volumes. The initial 2D annotated slices are then used to train a 2D model and infer segmentations of all other slices in the dataset. The predicted masks are then treated as a 3D segmentation and a 3D model. Finally, the annotator adjusts 3D model's predicted masks if necessary, and the resulting predictions are used for training a refined 3D model.

Their strategy is evaluated on the task of 3D body composition segmentation - skeletal muscle (SM), visceral adipose tissue (VAT), subcutaneous adipose tissue (SAT), and inter/intra-muscular adipose tissue (IMAT). They used an internal train (116 scans) and test (20 scans) set, and an additional external test set for visual evaluation (KiTS21 dataset). The visual evaluation process presented the predicted segmentation to three radiologists who used a 5-point scale to describe the manual correction effort that the predictions would need.

The paper demonstrates that high results were achieved with only 417 initially annotated slices. Their 2D to 3D strategy is clever and simple, and could be very useful in cases where segmentations are very dense and require a lot of effort to be done in 3D manually.

**Strengths:**

- As more and more foundation/pre-trained models are available, strategies such as these will be extremely useful for adapting them to other tasks
- The authors address a case where segmentations are very dense and labor-intensive to define in a whole 3D image. Their method is a simple yet effective way of leveraging 2D annotations to build 3D models.

**Weaknesses:**

- Would identifying and refining the 2D model's prediction and training one 3D model instead of two be sufficient? On the other hand, this might not be as important of a question - I assume that the time to train the 3D model was relatively short.
- Writing can be improved

**Detailed Comments:**

- The abstract can be rewritten with a better flow, it doesn't do justice to the paper.
- SM, VAT, and SAT abbreviations not defined earlier in the abstract
- Several missing citations "(?)"- pages 2, 8
- Page 2, sentence "Resulting in a model achieving high Dice scores, however, ..." can be improved. There are several other such sentences.

**Justification Of The Preliminary Rating:**

The proposed method is sound. The task of body composition is appropriate for the problem that the proposed method addresses. I only found minor problems in writing and had a few questions out of interest rather than out of concern.

**Questions To Address In The Rebuttal:**

- Minor question: how was this implemented? Was something like MONAI Label used or were the steps done in more of a manual way?

---

> ### Author Response · Authors · 2024-03-18
>
> We thank the reviewers for their constructive feedback and the recognition of the strengths in our work.
> We agree that there is room for improvement in writing and communication. We have taken your suggestions into consideration and have revised the abstract for better flow and clarity. Additionally, we have meticulously reviewed the document to address and correct missing citations and improve sentence structures.
>
> Regarding the question of refining the 2D model's predictions and potentially training one 3D model instead of two, our approach was driven by a desire for granularity in model performance. The decision to train two models allowed for more focused optimization on this specific segmentation challenges, thus enhancing overall accuracy.  However, we acknowledge that consolidating training into a single 2D model and a single 3D model could offer advantages in terms of training time and resource allocation, making it a valuable point for future exploration.
>
> ### *Minor question: how was this implemented? Was something like MONAI Label used or were the steps done in more of a manual way?*
>
> For the implementation, we trained the 2D model using our own U-Net architecture and pipeline, granting us flexibility for fine-tuning and exploring strategies like adjusting regularization and layer modifications. Ultimately, we achieved best results with a simple training approach, that could easily be implemented using MONAI. The 3D model was trained with the nnUNet framework, using its default settings without additional modifications.

---

### Official Review · Reviewer_waku · 2024-03-04

**Confidence:** 3
**Preliminary Rating:** 3
**Final Rating:** 3.5

**Summary:**

This research introduces an annotation-efficient strategy for 3D body composition segmentation using CT images, addressing challenges in manual annotations. The study employs an iterative self-learning approach, transitioning from a 2D to a 3D model, significantly reducing manual annotation efforts. The final model demonstrates excellent segmentation accuracy for skeletal muscle, adipose tissues, and achieves high Dice scores. The significance lies in offering a robust tool for precise body composition analysis, enhancing diagnostic and prognostic assessments in clinical settings, with potential applications in fields such as oncology.

**Strengths:**

The paper demonstrates notable strengths in addressing the challenges of manual annotations in 3D body composition segmentation using CT images. The iterative self-learning strategy, transitioning from 2D to 3D models, is a novel and efficient approach, showcasing innovation in methodology. Despite potential limitations in benchmark performance, the paper's scientific merit lies in its potential value to the medical community, offering a robust tool for accurate body composition analysis. The clear structure, language, and adherence to scientific principles contribute to the paper's overall strength, making it a valuable contribution to the field. I would rate the paper as highly valuable given its innovative methodology and potential impact on clinical assessments.

**Weaknesses:**

One potential weakness of the paper lies in the limited size of the internal test set (20 cases), raising concerns about the generalizability of the proposed model. Additionally, the lack of a comprehensive comparison with existing state-of-the-art methods in the field may impact the paper's ability to establish its superiority. While the iterative self-learning strategy is innovative, the paper could benefit from a more thorough exploration of potential limitations and challenges associated with this approach. Further, the absence of a detailed discussion on the potential impact of imaging artifacts and variability in scan qualities on the model's performance is a notable gap. Addressing these aspects would enhance the paper's completeness and credibility.

**Detailed Comments:**

• Clarify the criteria used for selecting the 20 cases in the internal test set to ensure transparency and allow readers to assess the representativeness of the evaluation.
• Provide a more extensive discussion on the limitations and challenges associated with the proposed iterative self-learning strategy, addressing potential drawbacks and areas for improvement.
• Include a more comprehensive review of related work, particularly focusing on existing methods for 3D body composition segmentation, to provide a clearer context for the novelty of the proposed approach.
• Consider discussing the potential implications of imaging artifacts, noise, and variations in scan qualities on the model's performance, offering a more thorough analysis of real-world applicability.
• If available, consider providing publicly accessible source code or datasets to facilitate reproducibility and allow the research community to further validate and build upon the proposed model.
• Consider expanding the discussion on the clinical implications of the proposed model, emphasizing how it can contribute to improving patient care and outcomes in real-world medical scenarios.

**Justification Of Final Rating:**

While the paper addresses the concerns raised in the review, the paper still does not have any comparisons to existing methods. Since both the dataset and method is new in the paper, it makes it impossible to comment of the effectiveness of the method when compared to previous methods and hence I maintain my rating but bump to borderline accept.

**Justification Of The Preliminary Rating:**

While the paper introduces an innovative iterative self-learning strategy for 3D body composition segmentation using CT images, potential concerns include the limited size of the internal test set (20 cases) and the absence of a comprehensive comparison with existing state-of-the-art methods. Additionally, the paper could benefit from a more in-depth exploration of the limitations associated with the proposed strategy and a more thorough discussion of the real-world implications of the model, particularly in the presence of imaging artifacts and varying scan qualities. Addressing these aspects in the rebuttal could potentially influence the rating.

**Questions To Address In The Rebuttal:**

In their rebuttal, it would be valuable for the authors to offer further details on the criteria guiding the selection of the 20 cases in the internal test set, ensuring transparency and addressing concerns of potential biases. A more in-depth discussion on the limitations and challenges associated with the proposed iterative self-learning strategy, along with concrete strategies for improvement, would strengthen the paper. Additionally, emphasizing the real-world implications of the model by discussing its performance in the presence of imaging artifacts, noise, and variations in scan qualities would contribute to a more comprehensive understanding of its robustness. To enhance the novelty of their approach, the authors are encouraged to provide a more thorough review of related work, particularly focusing on existing methods for 3D body composition segmentation. Addressing these aspects in the rebuttal would positively impact the paper's clarity, credibility, and potential impact on the scientific community.

---

> ### Author Response · Authors · 2024-03-18
>
> We thank the reviewers for their insightful comments on our paper, which have highlighted its strong aspects and, more crucially, the sections that demand further clarification and improvement which helped us to increase the quality of the manuscript. We address each of their points as follows:
>
> ### *In their rebuttal, it would be valuable for the authors to offer further details on the criteria guiding the selection of the 20 cases in the internal test set, ensuring transparency and addressing concerns of potential biases.*
>
> We agree with the reviewer on the importance of ensuring transparency, thoroughly investigating potential biases, and reporting them.
> We selected our samples based on the field of view, focusing on opportunistic scans from patients undergoing routine CT procedures in clinical practice. Our sampling was stratified by age and sex to ensure diversity within our study population. However, we regret to inform that we cannot supply additional information regarding BMI or specific diagnoses for these individuals, as this information is not so easy to extract due to the anonymisation process. In our future work, we will also try to obtain this information in order to further increase the interpretability of the results. For our external test set, we chose 100 cases out of 300 available in the KiTS dataset, again prioritising the field of view (L1 - L5) and ensuring stratification by BMI, age, and sex to foster a comprehensive analysis.
> To aid in the understanding of our dataset composition, we have included a table in the appendix that provides an overview of the sex, age, and acquisition parameters for our internal test and training sets, as well as a separate overview for the external test set used in the reader study, which additionally includes BMI data. We believe this detailed breakdown will offer clarity and support the validity of our findings.
>
> ### *A more in-depth discussion on the limitations and challenges associated with the proposed iterative self-learning strategy, along with concrete strategies for improvement, would strengthen the paper.*
>
> We included a paragraph in the discussion about challenges associated with self-learning (such as the risk of overfitting to noise) and suggest future improvements (such as uncertainty assessments and exploring strategies like modifying loss functions and sample weighting to mitigate negative impacts on learning).
>
> ### *To enhance the novelty of their approach, the authors are encouraged to provide a more thorough review of related work, particularly focusing on existing methods for 3D body composition segmentation.*
>
> To provide a more thorough overview of related work, we included additional references for studies focusing on single slice segmentation (for L3, L3/L4, pelvis, multiple heights, end-to-end solutions). Regarding existing methods for 3D body composition, we added two references for non-DL methods and three references for DL models (Liu et al., 2020), (Fu et al., 2020), (Dai et al., 2024). We focused on DL approaches for muscle and adipose tissue segmentations using CT scans and excluded methods that only segmented adipose tissue or muscles and methods that work for MR images.
>
> ### *Additionally, emphasizing the real-world implications of the model by discussing its performance in the presence of imaging artifacts, noise, and variations in scan qualities would contribute to a more comprehensive understanding of its robustness.*
>
> To enhance our model's robustness across diverse clinical settings and scan protocol variances, we incorporated scans from various manufacturers and acquisition parameters. Our training set specifically included low-dose scans and contrast-enhanced scans, alongside a broad spectrum of kernel types that introduce different levels of noise, all aimed at improving the model's adaptability. Based on this, we expect a robust performance, however this remains to be investigated in a larger validation study with additional external data. We included an overview of failure cases in the appendix F. This section illustrates challenges stemming from artifacts, truncation, as well as conditions such as dilated bowel structures and fluid in subcutaneous adipose tissue.
>
> ### *Consider expanding the discussion on the clinical implications of the proposed model, emphasizing how it can contribute to improving patient care and outcomes in real-world medical scenarios.*
>
> We propose to add this to the conclusion (enabling precise monitoring of changes, providing deeper insights into patients’ health status, assessing these measurements holds potential for refining outcome predictions and diagnostics)
>
> ### *If available, consider providing publicly accessible source code or datasets to facilitate reproducibility and allow the research community to further validate and build upon the proposed model.*
>
> We are planning on making the trained model available on grand-challenge up on request for inference.

---

### Comment · Area_Chair_QkVo · 2024-03-19
**The discussion period begins**

Dear reviewers and authors,

Thank you for your contribution to MIDL24. The discussion period begins! I encourage all reviewers and authors to participate in the discussion to address questions and clarify uncertainties.

Thank you!

---

### Meta-Review · Area_Chair_QkVo · 2024-04-03

**Recommendation:** Accept (Poster)
**Confidence:** 5

**Metareview:**

Three reviewers intend to accept this work, although, after the rebuttal, the authors provide no comparison to existing methods, which I think is a drawback of the paper. The iterative learning is a commonly used strategy for reducing annotation efforts in medical imaging. In my view, the novelty of this specific iterative learning strategy is to use 2D and 3D UNet methods together in the annotation process.

Because muscle and adipose tissue segmentation is a relatively new field, and the experiment includes a multi-reader study, this paper can provide more insights for other researchers to develop similar tools or datasets. Thus, I align with the reviewers’ decision to accept this paper. I recommend that in the future the author could enhance their contribution by comparing their work with other existing work such as [1].

[1] Blankemeier, Louis, Arjun Desai, Juan Manuel Zambrano Chaves, Andrew Wentland, Sally Yao, Eduardo Reis, Malte Jensen et al. "Comp2comp: Open-source body composition assessment on computed tomography." arXiv preprint arXiv:2302.06568 (2023).

---

### Decision · Program_Chairs · 2024-04-05

Accept (Poster)